# S100A8 and S100A12 Proteins as Biomarkers of High Disease Activity in Patients with Rheumatoid Arthritis That Can Be Regulated by Epigenetic Drugs

**DOI:** 10.3390/ijms24010710

**Published:** 2022-12-31

**Authors:** Leszek Roszkowski, Bożena Jaszczyk, Magdalena Plebańczyk, Marzena Ciechomska

**Affiliations:** 1Department of Outpatient Clinics, National Institute of Geriatrics, Rheumatology and Rehabilitation (NIGRiR), 02-637 Warsaw, Poland; 2Department of Pathophysiology and Immunology, National Institute of Geriatrics, Rheumatology and Rehabilitation (NIGRiR), 02-637 Warsaw, Poland

**Keywords:** rheumatoid arthritis, biomarkers, epigenetics, S100 proteins, monocytes, DNA methylation, transcriptomics

## Abstract

Rheumatoid arthritis (RA) is an autoimmune chronic inflammatory disease that is still not well understood in terms of its pathogenesis and presents diagnostic and therapeutic challenges. Monocytes are key players in initiating and maintaining inflammation through the production of pro-inflammatory cytokines and S100 proteins in RA. This study aimed to test a specific DNA methylation inhibitor (RG108) and activator (budesonide) in the regulation of pro-inflammatory mediators—especially the S100 proteins. We also searched for new biomarkers of high disease activity in RA patients. RNA sequencing analysis of healthy controls (HCs) and RA monocytes was performed. Genes such as the S100 family, TNF, and IL-8 were validated by qRT-PCR following DNA-methylation-targeted drug treatment in a monocytic THP-1 cell line. The concentrations of the S100A8, S100A11, and S100A12 proteins in the sera and synovial fluids of RA patients were tested and correlated with clinical parameters. We demonstrated that RA monocytes had significantly increased levels of S100A8, S100A9, S100A11, S100A12, MYD88, JAK3, and IQGAP1 and decreased levels of IL10RA and TGIF1 transcripts. In addition, stimulation of THP-1 cells with budesonide statistically reduced the expression of the S100 family, IL-8, and TNF genes. In contrast, THP-1 cells treated with RG108 had increased levels of the S100 family and TNF genes. We also revealed a significant upregulation of S100A8, S100A11, and S100A12 in RA patients, especially in early RA compared to HC sera. In addition, protein levels of S100A8, S100A11, and S100A12 in RA synovial fluids compared to HC sera were significantly increased. Overall, our data suggest that the S100A8 and S100A12 proteins are strongly elevated during ongoing inflammation, so they could be used as a better biomarker of disease activity than CRP. Interestingly, epigenetic drugs can regulate these S100 proteins, suggesting their potential use in targeting RA inflammation.

## 1. Introduction

Rheumatoid arthritis (RA) is a chronic, inflammatory autoimmune joint disease that affects approximately 1% of the world’s population and is influenced by many genetic, epigenetic, and environmental factors. RA is a systemic disease characterized primarily by chronic arthritis that, if not effectively treated, results in joint damage and loss of function, followed by deterioration of the patient’s physical and social functions and disability [1]. The mechanism of the formation and development of RA is very complex, with monocytes and macrophages playing a central role. Monocytes contribute strongly to the progression of RA by producing pro-inflammatory molecules and infiltrating the inflamed synovium, resulting in tissue destruction [2]. These cells are involved in the secretion of key pro-inflammatory mediators, including the S100 family (e.g., S100A8, S100A9, S100A11, S100A12), cytokines (e.g., TNF, IL-1β, IL-6, GM-CSF), chemotactic agents (e.g., CCL2, CCL3, CCL5, CX3CL1, IL-8), and metalloproteinases (e.g., MMP-3 and MMP-12) [3,4,5]. Therefore, molecules that block the increased inflammatory properties of monocytes in RA are needed, along with new biomarkers of high disease activity.

A method that has recently given rise to new drugs is epigenetic modification, which plays an important role in the cellular programming of gene expression. Epigenetics is defined as reversible changes in gene function that do not involve changes in the DNA sequence [6]. Inflammation is related to epigenetics and plays a vital role in the innate immune system [7]. Epigenetic mechanisms are sensitive to external stimuli as well as drugs; therefore, they mediate gene–environment interactions. DNA methylation is one of the most studied epigenetic modifications—a process by which methyl groups are added to the cytosine at the 5’ position by DNA methyltransferases (DNMTs). Insertion of a methyl group into a DNA sequence modifies the chromatin structure and, ultimately, silences the gene [8]. Studies have found several drugs that are inhibitors of DNA methylation (e.g., RG108 [9,10,11], azacitidine [12], decitabine [13], zebularine [14]) and a drug that is an activator of DNA methylation (budesonide [15,16]). These DNA methylation inhibitors and activator have already been successfully tested in research on the treatment of various types of cancer and rheumatic diseases, including RA [17].

In particular, RG108 {2-(1,3-dioxoisoindolin-2-yl)-3-(1H-indol-3-yl) propanoic acid or N-Phthalyl-L-tryptophan} is a compound specifically designed to block the active site of DNMTs [18]. Its proven ability to reactivate several tumor-suppressor genes and its lack of cellular toxicity make RG108 a good candidate for epigenetic modulation therapies [9,10,19,20,21,22,23]. Budesonide belongs to the group of glucocorticoids and is a commonly used drug in many autoimmune diseases. Its effectiveness in reducing inflammation has been well known in medicine for a long time, and its use has recently been extended to COVID-19 as well [24,25]. Importantly, one of the mechanisms of action of budesonide is the increase in DNA methylation [15,26,27,28,29].

Knowing the importance of early detection and monitoring in RA treatment, our research focused on proteins from the S100 family. These proteins play a critical role in inflammation because they can activate the innate immune pathway sensed by receptors for advanced glycation end products (RAGEs) or Toll-like receptors (TLRs) [30,31]. Moreover, it has also been proven that S100A8 and S100A9 in particular induce the production of TNF, IL-6, IL-1β, and IL-8 in monocytes. These mediators are very important in initiating and maintaining inflammation [32]. Studies have shown that the levels of S100 proteins in sera and synovial fluid (SF) were significantly increased in RA patients, and their levels were correlated positively with disease activity [33,34]. The levels of these proteins are considered to be a potential biomarker in RA because they are closely correlated with the radiographic progression of joint damage, ultrasound synovitis, and therapeutic response [35,36,37]. However, none of the previous studies made a distinction between patients with early RA (eRA) and advanced RA (aRA). Therefore, we wanted to check the levels of the S100 family proteins in eRA and aRA sera and synovial fluids compared to HCs. Furthermore, in our work, we wanted to verify whether epigenetic drugs such as RG108 and budesonide could modulate proteins from the S100 family. We also searched for new, even better biomarkers that would precisely correlate with disease activity and that could be modified by epigenetic drugs.

## 2. Results

### 2.1. Different Expressions of Pro-Inflammatory and Anti-Inflammatory Genes in Monocytes from RA Patients Based on Global Profiling

First, the global transcriptomic profiles of RA monocytes (*n* = 4) and age- and sex-matched HCs (*n* = 4) were analyzed to better understand the mechanisms of immune dysregulation in RA patients. The characteristics of RA patients in all experiments are presented in Table 1. RA patients selected for transcriptomic analysis were positive for RF and anti-CCP antibodies, and all of these RA patients were treated exclusively with NSAIDs and DMARDs to eliminate the potential influence of steroid and biological treatments on their immune pathways. The transcriptomic data from RNA-Seq were examined using a volcano plot, which revealed seven genes (red dots) that were highly upregulated in RA monocytes and two genes (blue dots) that were significantly downregulated in RA monocytes compared to HCs (Figure 1A). We also calculated the fold change of S100A8, S100A9, JAK3, S100A12, IQGAP1, MYD88, S100A11, IL10RA, and TGIF1 transcripts between RA and HCs, illustrated as a heatmap (Figure 1B). We found increases in the concentrations of S100A8 (2.2-fold, *p* = 0.024), S100A9 (1.9-fold, *p* = 0.017), S100A11 (1.2-fold, *p* = 0.007), S100A12 (1.6-fold, *p* = 0.007), MYD88 (1.2-fold, *p* = 0.008), JAK3 (1.8-fold, *p* = 0.009), and IQGAP1 (1.4-fold, *p* = 0.02) in patients with RA compared to the HCs. In contrast, we found reductions in fold change in RA patients relative to the control group in IL10RA (0.7-fold, *p* = 0.011) and TGIF1 (0.3-fold, *p* = 0.046) (Figure 1B). Overall, these data strongly suggest an important role for S100 family protein dysregulation in RA monocyte transcriptomic data.

### 2.2. Viability of the Monocytic THP-1 Cell Line Treated with Budesonide and RG108

The viability of the monocytic THP-1 cell line was determined by chemiluminescence and by the MTS test after treatment with various concentrations of budesonide (DNA methylation activator) and RG108 (DNA methylation inhibitor) at different time points (24 h, 48 h, and 72 h) (Appendix A). We used graded concentrations of drugs and selected 10 µM budesonide and 100 µM RG108, since these high concentrations did not lead to significant cell death due to their toxicity, but at the same time the drugs were able to modify DNA methylation based on previous literature [22,27]. 

### 2.3. Different Expressions of Pro-Inflammatory and Anti-Inflammatory Genes

Next, we examined genes, with particular emphasis on genes from the S100 family, based on qPCR analysis after 24 h, 48 h, and 72 h of incubation with drugs (i.e., budesonide or RG108). We selected 48 h drug stimulation, which was not toxic to the monocytic THP-1 cells, and the obtained results were consistent with the expected trends (Appendix A). Then, the THP-1 cells were treated with budesonide or RG108 in the presence of LPS to create an inflammatory environment. We found that the expression levels of S100A8 (3.0-fold, *p* = 0.0390), S100A9 (4.4-fold, *p* = 0.0317), S100A12 (3.3-fold, *p* = 0.0260), IL-8 (6.8-fold, *p* = 0.0286), and TNF (7.5-fold, *p* = 0.0260) were significantly decreased after treatment with budesonide compared to untreated THP-1 cells (Figure 2A–F). To better illustrate the fold change in the expression of pro-inflammatory genes between untreated + LPS THP-1 cells and THP-1 cells treated with budesonide + LPS, a heatmap was introduced (Figure 2G). On the other hand, it can be seen that the expression levels of S100A8, S100A9, S100A12, and TNF were increased after RG108 treatment compared to untreated THP-1 cells (Figure 2A–F). Even though statistical significance was not achieved, an evident trend is visible, suggesting that RG108 is an inhibitor of DNA methylation and increases the expression of many genes, including genes from the S100 family.

### 2.4. Secretion of S100 Family Proteins in Sera and Synovial Fluids

Based on previously published data highlighting the important role of the S100 family in RA, we measured the serum levels of the S100A8, S100A11, and S100A12 proteins in RA patients and also divided them into two groups—eRA and aRA—compared to HC serum levels. In the entire group of RA patients, we observed statistically significantly increased levels of S100A8 (*p* = 0.0116), S100A11 (*p* = 0.0138), and S100A12 (*p* = 0.0296) proteins compared to HCs. Additionally, in eRA sera, the levels of S100A8 (*p* = 0.0024), S100A11 (*p* = 0.0020), and S100A12 (*p* = 0.0059) were statistically significantly increased (Figure 3A–C). We also tested the concentrations of S100A8, S100A11, and S100A12 proteins in aRA patients in the SF. Protein levels of S100A8 (*p* = 0.0005), S100A11 (*p* = 0.0007), and S100A12 (*p* = 0.0005) in aRA SF compared to HC sera were significantly increased (Figure 3A–C). We did not measure the contents of S100A9 because S100A8 and S100A9 exist as a heterodimeric complex called calprotectin (CLP). Indeed, in this complex, S100A8 is predominantly responsible for CLP activity; therefore, we only measured the S100A8 protein in HC and RA sera and synovial fluid [38].

In order to select the best biomarker indicating high disease activity in RA patients, we analyzed several clinical indicators that could be used to verify disease activity. In our study, S100A8 was positively correlated with all parameters: CRP (*p* = 0.01), DAS28-CRP (*p* = 0.01), ESR (*p* = 0.002), and DAS28-ESR (*p* = 0.002) (Figure 4A,D,G,J). Moreover, S100A12 was positively correlated with DAS28-CRP (0.007), ESR (*p* = 0.04), and DAS28-ESR (*p* = 0.003) (Figure 4F,I,L). Additionally, the concentration of S100A11 was positively correlated only with ESR (*p* = 0.01) (Figure 4H). These data suggest that the proteins S100A8 and S100A12 in particular are significantly increased during ongoing inflammation, so they can be used as biomarkers of high disease activity.

Finally, to assess the diagnostic potential of S100A8, S100A12, and S100A11 in sera as a useful biomarker of high disease activity (DAS28 > 5.1) in patients with RA, receiver operating characteristic (ROC) curve analysis was performed. In Figure 5A,B, it can be seen that the S100A12 protein in sera was much more specific (area under the curve (AUC) = 0.87, *p* = 0.0006; and AUC = 0.83, *p* = 0.002, respectively) in all RA and eRA patients than the CRP level (AUC = 0,63, *p* = 0.215; and AUC = 0.52, *p* = 0,874). These results indicate that S100A12 can serve as a better biomarker of high disease activity than the CRP level. Moreover, the S100A8 protein in sera was significantly more specific (AUC = 0.78, *p* = 0.006) in all RA patients than the CRP level (AUC = 0.63, *p* = 0.215), which also indicates that S100A8 was superior to CRP as a biomarker of high disease activity (Figure 5A). However, CRP reached statistical significance only in the aRA group (AUC = 0.89, *p* = 0.033) (Figure 5C)

### 2.5. PPI Network Analysis and Biological Function Annotation

In order to find the functional effects of four selected S100 family genes related to the transcriptome of RA monocytes, we used the STRING database. The network contained the genes *S100A8, S100A9, and S100A12*—with very strong links and interactions between them—and S100A11, with less interaction (Figure 6A). All biological processes for these genes were related to acute-phase response, neutrophil chemotaxis and degranulation, and regulation of the inflammatory response. Functional annotations for the Reactome pathways of these genes were obtained with the g:Profiler program. In Figure 6B, it can be seen that these genes play a key role in the innate immune system, metal sequestration by antimicrobial proteins, Toll-like receptor (e.g., TLR1, TLR2, TLR4, TLR6) cascade, the MyD88:MAL cascade initiated on plasma membranes, neutrophil degranulation, and MyD88 and IRAK4 deficiency.

## 3. Discussion

In this study, using RNA-Seq, we have analyzed the transcriptomic profiles of monocytes from RA patients compared with HCs. We identified pro-inflammatory genes that are highly elevated in RA monocytes, which play an important role in the progression of RA. Then, we examined the effects of drugs that are known as DNA methylation inhibitors (RG108) and activators (budesonide) on monocytic THP-1 cells, with particular emphasis on genes from the S100 family. Additionally, we found statistically significant elevated levels of S100A8, S100A11, and S100A12 proteins in both the sera and SF of RA patients, as well as their correlation with clinical data.

We demonstrated that the expression of the S100 family and other inflammation-related genes was significantly upregulated in RA compared to HC monocytes. This confirmed that the pro-inflammatory transcripts of the S100 family—such as S100A8, S100A9, S100A11, and S100A12—are much more abundant in the monocytes of RA patients, which is consistent with other findings [39]. Along with our results shown in Figure 1, previous studies demonstrated elevated levels of other pro-inflammatory genes, such as JAK3, IQGAP1, and MYD88 [40,41,42]. In contrast, anti-inflammatory genes such as TGF1 and IL10RA, for example, have been shown to be significantly reduced [43,44].

Many studies suggest that DNA methylation may contribute to the development or treatment of rheumatic diseases, including RA, so we have conducted studies with drugs with proven epigenetic effects [17]. Budesonide is a well-known corticosteroid anti-inflammatory drug that is widely used to relieve local inflammatory symptoms, e.g., in inflammatory bowel and respiratory diseases. Importantly, it is insufficiently absorbed due to the extensive metabolism of first-pass enzymes in the liver by cytochrome P450 3A4, which consequently has a very low oral bioavailability in humans and, therefore, is not used in the treatment of RA [45,46]. Fortunately, core–shell nanocarriers have been developed to deliver budesonide in the treatment of RA, which will allow us to overcome this problem in the future [47,48]. Moreover, budesonide has been shown to increase DNA methylation [15], which is why we used this drug in our study. Indeed, we confirmed that in the LPS-induced inflammation environment following treatment with budesonide, there were significant reductions in the levels of pro-inflammatory genes such as S100A8, S100A9, S100A12, IL-8, and TNF. These results are consistent with the findings of other studies demonstrating that budesonide also acts through the mechanism of extensive hypermethylation in DNA [26,28]. The second drug that we chose was a novel non-nucleoside small-molecule DNMT inhibitor—RG108. This drug has low genotoxicity and cytotoxicity compared to other DNMT inhibitors because it is more selective. Moreover, it does not direct demethylation of minor satellite repeats, indicating that RG108 administration will not result in genomic instability of the cell and potential mutagenesis [49]. Several studies have used RG108 to investigate epigenetic mechanisms related to physiological conditions, as well as cancers, including human prostate cancer cells, breast cancer cells, or colon cancer cells, as well as neurological diseases [9,10,11,50,51]. Furthermore, the data show that RG108 can ameliorate the cellular damage caused by oxidative stress by restoring an altered methylation pattern of aging genes, resulting in better treatment efficacy in diseases related to aging or inflammation [22]. Unlike older DNA methylation inhibitors such as azacitidine, decitabine, and zebularine, RG108 has not been studied in relation to rheumatic diseases and RA. Although we did not achieve statistical significance, we found a trend towards increased expression of pro-inflammatory genes upon RG108 treatment.

In the next part of our study, following the transcriptome analysis, we selected the proteins S100A8, S100A11, and S100A12 to test whether these proteins could be used as biomarkers of RA activity. In RA, the most studied protein from the S100 family is CLP, which is a heterodimeric complex of S100A8 and S100A9. Importantly, S100A8 is responsible for the activity of the CLP complex, while S100A9 only protects S100A8 from degradation, which is why we only selected S100A8 in our study [38]. Numerous studies have shown that the proteins that we chose play a fundamental role in the pathogenesis and inflammatory response in RA [34]. Furthermore, S100 proteins are known as biomarkers of inflammation in many diseases, including psoriatic arthritis, juvenile idiopathic arthritis, and cystic fibrosis [52,53,54]. We confirmed the increased concentration of the S100A8 protein in the sera of RA patients relative to HCs. Interestingly, this concentration was statistically significant in the eRA group (disease duration of less than a year), which was probably due to the intensive and prolonged immunosuppressive treatment of patients with aRA where there is only an elevated trend. We were the first to identify this because, in previous studies, researchers did not divide patients into eRA and aRA groups like we did. It should also be noted that patients in the eRA group, in addition to taking fewer DMARDs and corticosteroids, were younger and had higher disease activity measured by DAS28-ESR and DAS28-CRP than patients in the aRA group, which may have influenced the results. Indeed, the level of S100A8 was significantly correlated with the clinical and laboratory assessments of joint inflammation in RA patients (such as CRP and DAS28-ESR), similar to the meta-analysis study [33]. Furthermore, it is important that S100A8 levels were a better diagnostic biomarker for high disease activity than CRP levels in RA patients, as has also been confirmed by previous studies [36,37]. Similarly, the levels of the S100A11 protein were significantly increased in patients with RA—and especially with eRA—compared to HCs, but positively correlated only with ESR. Consistent results were also obtained by Cerezo et al., but they did not achieve a statistically significant difference, probably due to the comparison of RA sera to osteoarthritis sera but not to HC sera as we did [55]. A significant increase in S100A11 was also observed in the SF of RA patients—as also confirmed in our study—which was related to the distribution of this protein [56]. Another protein that plays an important role in the pathogenesis and maintenance of RA is S100A12 [57]. In our study, we found statistically increased concentrations of the protein S100A12 in the sera of RA patients compared to HCs. These results are consistent with a previous work by Foel et al. [58]. Moreover, they proved that the response to therapy was accompanied by a marked decrease in serum S100A12 levels after RA treatment [58]. Indirectly, we also observed this in our work, where higher concentrations of the S100A12, S100A8, and S100A11 proteins occurred in eRA than in aRA, which is a new finding that has not been previously emphasized. The level of S100A12 was positively correlated with clinical parameters such as DAS28-CRP, ESR, and DAS28-ESR, and was a better diagnostic biomarker of high disease activity than CRP levels in all RA patients and eRA. This suggests that the level of S100A12 protein in sera can be used as a non-invasive diagnostic biomarker reflecting strongly upregulated expression of S100A12 in inflamed joints [59,60]. Only in the aRA group did CRP in the ROC analysis turn out to be a better biomarker of high disease activity than S100A12, which is a limitation related to a small group of aRA patients with high disease activity due to their treatment. Additionally, we found high concentrations of the S100A8, S100A11, and S100A12 proteins in the SF, which were also proven in other studies to be good biomarkers of RA in the SF [55,57]. Moreover, using the STRING interactive network viewer, which can cluster networks on demand, demonstrated strong protein–protein interactions between S1000A8, S100A9, and S100A12, but fewer with S100A11 (Figure 6A) [5]. Furthermore, functional enrichment analysis using the g:Profiler tool confirmed strong involvement of the S100A8, S100A9, and S1000A12 proteins in the pathways responsible for the innate immune system and neutrophil degranulation (Figure 6B). In the case of other pathways particularly related to the TLR cascade, the contribution of the S100A11 protein is much smaller compared to that of the other proteins tested [39]. This observation supports our results, indicating that the S100A11 protein plays a minor role as a biomarker of inflammatory RA compared to S100A8 and S100A12 (Figure 5).

In conclusion, our study showed that the S100A8, S1000A11, and S1000A12 protein levels were increased in RA monocytes similarly to their enhanced production in serum and SF in RA patients. In conclusion, our study showed that S100A8, S1000A11, and S1000A12 transcripts were all increased in RA monocytes, and their secretory proteins were enhanced in the sera and SF of RA patients compared to HC sera. It should be noted that there are some limitations in our study, including the relatively small subgroups of patients, although we did achieve statistical significance of protein concentrations, as confirmed by linear regression and ROC analyses. It is also difficult to compare groups of older patients treated for many years for RA (aRA) with a group of often-younger patients not treated for RA (eRA). In our clinical practice, patients with aRA without immunosuppressant treatment were not observed, whereas 40% of eRA patients did not receive any immunosuppressants, as shown in Table 1. Our data suggest that the S100A8 and S100A12 protein levels are strongly increased during ongoing inflammation—especially in eRA, where the increase was more potent than in aRA—and correlate well with disease activity (DAS28-ESR, DAS28-CRP), so they can be used as biomarkers of disease activity. Interestingly, the proteins S100A8, S100A9, and S100A12, which were dysregulated in RA monocytes based on transcript data, can be regulated by epigenetic drugs such as budesonide and RG108, suggesting their potential use in targeting inflammation in RA.

## 4. Materials and Methods

### 4.1. Participants and Cell Purification

In this study, we used blood from 43 patients who met the 2010 American College of Rheumatology/European League Against Rheumatism classification criteria for RA [61]. The RA patients included 20 patients with eRA (i.e., disease duration less than one year) and 23 patients with aRA (i.e., disease duration greater than 2 years) (Table 1). All subjects gave their informed consent for inclusion before they participated in the study. The study was conducted in accordance with the Declaration of Helsinki, and the protocol was approved by the Ethics Committee of the National Institute of Geriatrics Rheumatology and Rehabilitation (NIGRiR) in Warsaw (approval no KBT-5/3/2019). Patients’ blood and SF were collected at the NIGRiR. A total of 19 healthy donors (17 women and 2 men) with no history of autoimmune disease were included as HCs; their median age was 39 years, and the range was 24–67. Blood from HCs was collected directly from healthy volunteers. Blood was collected in EDTA-coated tubes from HCs and RA patients during standard outpatient procedures. Monocytes were isolated from peripheral blood mononuclear cells (PBMCs) according to the manufacturer’s protocol using the CD14 + MACS beads isolation kit (Miltenyi Biotec, the Netherlands), as described in [62,63]. Purified monocytes were removed from the column and checked for purity by flow cytometry (showing > 95% purity for CD14^+^ CD3^−^ cells). Serum samples from HCs and RA patients were collected in serum separation tubes (BD Vacutainer SST II Plus), centrifuged for 10 min at 1500 g and 4 °C, aliquoted, and then frozen to −80 °C. The rheumatologist collected SF from four aRA patients with knee inflammation via a sterile knee puncture using 20 G needles. The SF was processed within 60 min of collection. Freshly aspirated SF (3.0 mL) was collected in the serum separation tubes. The fluid was centrifuged for 10 min at 1500 g and 4 °C and then frozen to −80 °C.

### 4.2. Transcriptomic Sequencing

Total RNA from monocytes was isolated according to the manufacturer’s protocol using the miRNA easy kit (Qiagen, Manchester, UK). The quality and integrity of the total RNA of 8 samples (4 HC and 4 RA) were assessed with an Agilent 2100 Bioanalyzer using the Agilent RNA 6000 Pico Chip (Agilent Technologies, Santa Clara, CA, USA) and reached RNA integrity scores ≥ 7. High-throughput next-generation sequencing (NGS) was carried out using the TruSeq Stranded Total RNA Library Prep kit (cal. No. RS-122–2201; Illumina, San Diego, CA, USA). Gene expression was profiled using an Illumina HiSeq 2000 sequencer. The results of RNA-Seq were then processed by selecting genes where the *t*-test score was less than 0.05 and FDR < 0.05, which was considered to be significant. Following the aforementioned selection method, 1091 genes were obtained and analyzed in terms of the available literature.

### 4.3. THP-1 Cell Culture

THP-1 monocytic cells were seeded in 24-well plates at a concentration of 0.5–1 × 10^6^ cells/mL and cultured for 24 h in 1 mL of RPMI (Laboratory of General Chemistry, IITD, Poland) supplemented with penicillin (100 U/mL), streptomycin (100 μg/mL), L-glutamine (2 mM), and 10% FBS (all from Life Technologies, Carlsbad, CA, USA) at 37 °C in 5% CO_2_. Cells were stimulated for 48 h with 1 μg/mL TLR4 agonist (LPS from *E. coli* O111:B4 InvivoGen, Toulouse, France), budesonide (B7777-250 MG Sigma-Aldrich, Darmstadt, Germany), and RG 108 (R8279-10 MG Sigma-Aldrich, Germany).

### 4.4. Cell Viability

After treatment with various concentrations of drugs (i.e., budesonide or RG108), the Cell Titer 96 Aqueous One Solution cell proliferation assay (MTS assay; Promega Corp., Madison, WI, USA) was used to quantify the relative viability of THP-1 cells. Assays were performed according to the manufacturer’s instructions (Appendix A). Another method was also used to determine the relative viability of THP-1 cells after administration of various concentrations of these drugs (i.e., budesonide, RG108). We used Cell-Titer-Glo Luminescent assays (Promega, Madison, WI, USA) according to the manufacturer’s protocol (Appendix A). We also checked the viability of the THP monocytic cell line as measured by the MTS assay (Promega Corp., Madison, WI, USA) after treatment with budesonide and RG 108, after 24 h, 48 h, and 72 h of incubation (Appendix A).

### 4.5. qPCR Analysis and Gene Expression

The expression levels of 10 genes—S100A8 (Hs00374264_g1), S100A9 (Hs00610058_m1), S100A11 (Hs01055944_g1), S100A12 (Hs00942835_g1), IL-8 (Hs00174103_m1), TNF (Hs00174128), IQGAP1 (Hs00896595_m1), TGIF1 (Hs00820148_g1), IL10RA (Hs00155485_m1), and MYD88 (Hs01573837_g1)—were assessed. These genes were previously selected from RNA-Seq analysis of RA monocytes. The tests were carried out on THP-1 cells after the addition of LPS and budesonide or RG108. For transcriptomic validation, cDNA was synthesized using a High-Capacity cDNA RTKit (Thermo Fisher Scientific, Waltham, MA, USA) and normalized to 18S as an internal control. The samples were analyzed in triplicate using QuantStudio 5 qRT-PCR machines (Thermo Fisher Scientific, Waltham, MA, USA). Gene expression levels relative to mean HC (arbitrarily set to 1) were calculated using the following equation: (2−△△CT) − 1.

### 4.6. Enzyme-Linked Immunosorbent Assays (ELISAs)

Sera were isolated from blood by routine laboratory methods, and samples were stored in aliquots at −80 °C until assayed. Serum and SF S100A8, S100A11, and S100A12 proteins levels were measured using commercially available ELISA kits, according to the manufacturer’s protocol—all from the CircuLex ELISA Kit (MBL a JSR Lite Sciences Company, Woburn, MA, USA).

### 4.7. Functional and Pathway Analysis

The genes S100A8, S100A9, S100A11, and S100A12, which form protein–protein interaction (PPI) networks that fulfill biological roles, were analyzed. The construction of the PPI networks was studied using the Search Tool for the Retrieval of Interacting Genes (STRING, v 11.5) (https://www.string-db.org/ (accessed on 12 August 2021)). Confidence scores > 0.7 were set as significant. STRING is a database that lists known or predicted protein interactions from high-throughput experiments, genomic analysis, conservative co-expression, and previously known literature [64]. We also used g:Profiler (https://biit.cs.ut.ee/gprofiler (accessed on 12 August 2021))—a collection of tools used in standard protein- or gene-focused computational analysis processes. Gene pathway analysis was generated with g:GOSt, which performs the functional gene enrichment analysis.

### 4.8. Statistical Analysis

All data are presented as the mean ± SEM. Group differences were tested for statistical significance using either the parametric two-tailed *t*-test or the Mann–Whitney test when the normality assumption was not met (GraphPad Software v7, Boston, MA, USA). A *p*-value < 0.05 was considered to be statistically significant; *p*-values were expressed as follows: ns for insignificant; 0.05 > *p* > 0.01 *; 0.01 > *p* > 0.001 **; *p* < 0.001 ***. To confirm the accuracy of the biomarkers, receiver operating characteristic (ROC) curve analysis was carried out. GraphPad Prism (GraphPad Software, USA) was used to calculate all tests. The cutoff value was based on DAS28-ESR > 5.1 (high disease activity), allowing us to show the maximum potential effectiveness of the biomarkers of high disease activity.

## Figures and Tables

**Figure 1 ijms-24-00710-f001:**
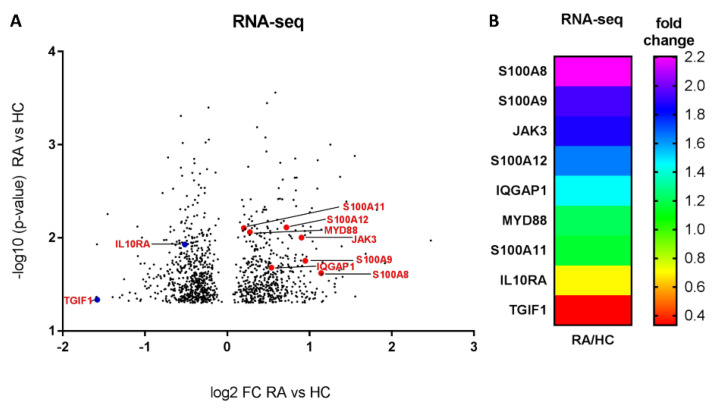
Global analysis of transcriptomic expression profiles between healthy control (HC) and rheumatoid arthritis (RA) monocytes. The volcano plot shows upregulated (red dots) and downregulated (blue dots) transcripts in RA (*n* = 4) compared with HC (*n* = 4) monocytes (**A**). Heatmap of S100A8, S100A9, JAK3, S100A12, IQGAP1, MYD88, S100A11, IL10RA, and TGIF1 transcripts in RA compared with HC monocytes. The colors in the heatmap represent the expression intensity of the transcripts (**B**).

**Figure 2 ijms-24-00710-f002:**
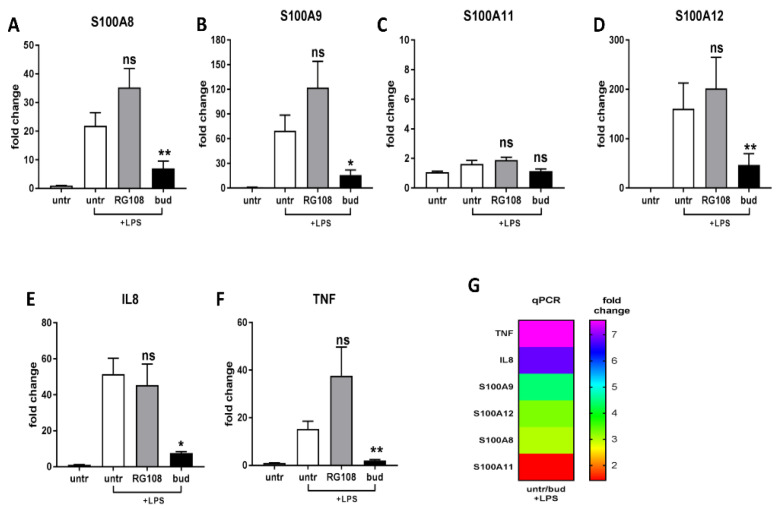
Comparison of the expression of selected genes in untreated THP-1 cells and those treated with budesonide or RG108 in the presence of LPS: The expression levels of S100A8 (**A**), A100A9 (**B**), S100A11 (**C**), S100A12 (**D**), IL-8 (**E**), and TNF (**F**) genes treated with budesonide or RG108 in the presence of LPS following 48 h incubation in THP-1 cells (*n* = 6). Heatmap of fold change in TNF, IL-8, S100A9, A100A12, S100A8, and S100A11 expression levels between untreated THP-1 cells and those treated with budesonide + LPS following 48 h of incubation, measured by qRT-PCR (*n* = 6) (**G**). *p* values were expressed as follows: 0.05 > *p* > 0.01 as *, 0.01 > *p* > 0.001 as ** and ns—not significant.

**Figure 3 ijms-24-00710-f003:**
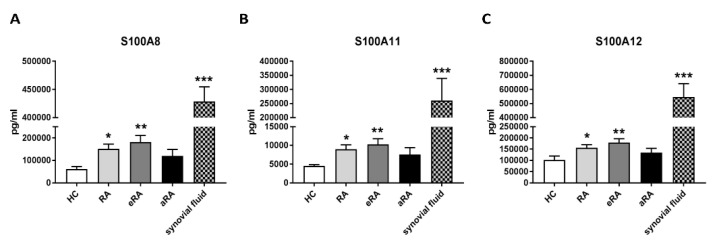
The levels of the S100A8, S100A11, and S100A12 proteins in the bodily fluids of RA patients: S100A8 (**A**), S100A11 (**B**), and S100A12 (**C**) protein concentrations in the sera of RA patients (*n* = 37–39), with early RA (eRA) (*n* = 18–20) and advanced RA (aRA) (*n* = 19), compared with HCs (*n* = 14–15). S100A8 (**A**), S100A11 (**B**), and S100A12 (**C**) protein concentrations were determined in the synovial fluid (SF) of RA patients (*n* = 4) compared to the serum concentrations of HCs (*n* = 14–15). *p* values were expressed as follows: 0.05 > *p* > 0.01 as *, 0.01 > *p* > 0.001 as **, 0.001 > *p* > 0.0001 as *** and ns—not significant.

**Figure 4 ijms-24-00710-f004:**
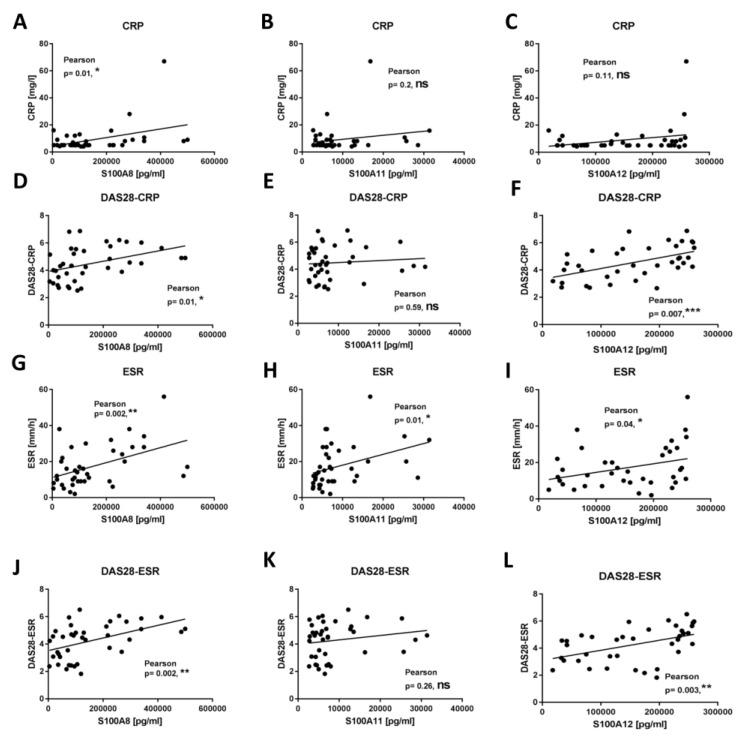
Correlation of S100A8, S100A11, and S100A12 proteins with clinical parameters: The levels of the S100A8 protein were measured in sera and correlated with clinical parameters—including CRP (**A**), DAS28-CRP (**D**), ESR (**G**), and DAS28-ESR (**J**)—in RA patients (*n* = 39). The levels of the S100A11 protein were measured in sera and correlated with clinical parameters—including CRP (**B**), DAS28-CRP (**E**), ESR (**H**), and DAS28-ESR (**K**)—in RA patients (*n* = 39). The levels of the S100A12 protein were measured in sera and correlated with clinical parameters—including CRP (**C**), DAS28-CRP (**F**), ESR (**I**), and DAS28-ESR (**L**)—in RA patients (*n* = 37). *p* values were expressed as follows: 0.05 > *p* > 0.01 as *, 0.01 > *p* > 0.001 as **, 0.001 > *p* > 0.0001 as *** and ns—not significant.

**Figure 5 ijms-24-00710-f005:**
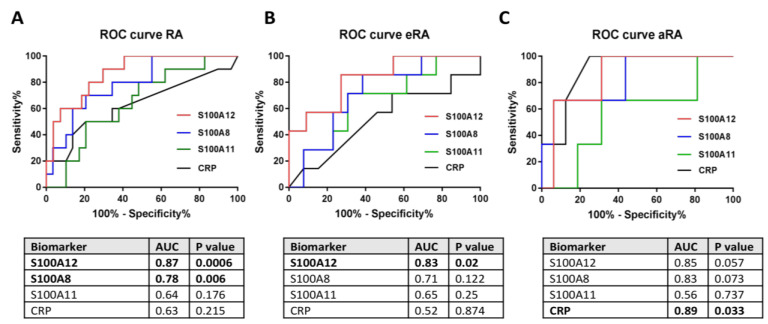
Receiver operating characteristic (ROC) curves for S100A8, S100A11, S100A12, and CRP in the sera of RA patients: ROC analysis comparing S100A12, S100A8, S100A11 proteins in sera vs. CRP levels in all RA patients (*n* = 37–39) (**A**); ROC analysis comparing S100A12, S100A8, and S100A11 proteins in sera vs. CRP levels in eRA patients (*n* = 18–20) (**B**); and ROC analysis comparing S100A12, S100A8, and S100A11 proteins in sera vs. CRP levels in aRA patients (*n* = 19) (**C**) as biomarkers of high disease activity. AUC: area under the curve.

**Figure 6 ijms-24-00710-f006:**
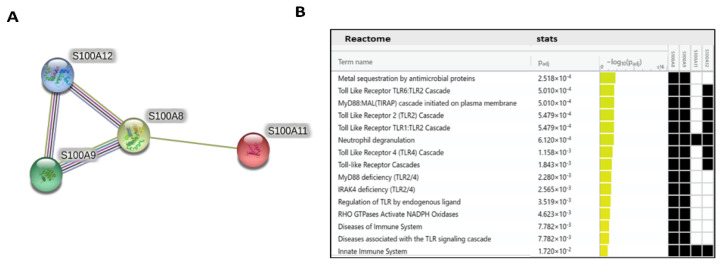
Network and Reactome analysis of selected genes from the S100 family: Association network analysis of the S100A8, S100A9, S100A11, and S100A12 genes was performed using the STRING program (**A**). Important pathways and processes in which the S100A8, S100A9, S100A11, and S100A12 genes were analyzed using the g:Profiler Reactome databases (**B**).

**Table 1 ijms-24-00710-t001:** Clinical and laboratory data of RA patients.

Parameters of RA Patients and HCs	Early RA (*n* = 20)	Advanced RA (*n* = 23)	All RA (*n* = 43)	HCs (*n* = 19)
Age, years, median (range)	45 (18–80)	62 (25–82)	58 (18–82)	42 (24–67)
Sex F/M	16/4	20/3	36/7	17/2
Disease duration, years, median (range)	0	9 (2–22)	2 (0–22)	N/A
Anti-CCP Abs, % (*n*)	75% (*n* = 15)	69% (*n* = 16)	72% (*n* = 31)	N/A
RF, % (*n*)	60% (*n* = 12)	60.9% (*n* = 14)	60.5% (*n* = 26)	N/A
CRP, mg/L, median (range)	6 (4–67)	5 (5–125)	5.5 (4–125)	N/A
ESR, mm/h, median (range)	15.5 (3–56)	17 (2–56)	16 (2–56)	N/A
DAS28-ESR, median (range)	4.66 (2.16–6.51)	4.32 (1.82–6.25)	4.56 (1.82–6.51)	N/A
DAS28-CRP, median (range)	4.675 (2.52–6.87)	4.33 (2.66–6.44)	4.51 (2.52–6.87)	N/A
Treatment				
Methotrexate, % (*n*)	45% (*n* = 9)	65.2% (*n* = 15)	55.8% (*n* = 24)	N/A
Leflunomide, % (*n*)	0% (*n* = 0)	4.3% (*n* = 1)	2.3% (*n* = 1)	N/A
Sulfasalazine, % (*n*)	20% (*n* = 4)	13% (*n* = 3)	16.3% (*n* = 7)	N/A
Hydroxychloroquine, % (*n*)	10% (*n* = 2)	13% (*n* = 3)	11.6% (*n* = 5)	N/A
Methylprednisolone, % (*n*)	15% (*n* = 3)	43.5% (*n* = 10)	30.2% (*n* = 13)	N/A
Without treatment, % (*n*)	40% (*n* = 8)	0% (*n* = 0)	18.6% (*n* = 8)	N/A

N/A—not applicable.

## Data Availability

The data presented in this study are available upon request from the corresponding author.

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
