# Peer review of "S100A8 and S100A12 Proteins as Biomarkers of High Disease Activity in Patients with Rheumatoid Arthritis That Can Be Regulated by Epigenetic Drugs"

_ijms, 2022, doi:10.3390/ijms24010710_

Round 1

Reviewer 1 Report

The authors have analyzed the expression of S100 family proteins in RA patients and healthy controls and its correlation with the clinical parameters. They also compared early RA and advanced RA patients. Furthermore, they studied if epigenetic drugs that regulate DNA methylation have any effect on S100 protein expression. They revealed a significant upregulation of S100A8, S100A11, and S100A12 in RA patients, especially in early RA compared to HC and the protein levels were also increased in SF. They suggest that S100A8 and S100A12 proteins are strongly elevated during ongoing inflammation, so these could be used as biomarkers of disease activity.  Epigenetic drugs influencing DNA methylation can modify S100 expression, suggesting their potential use in targeting inflammation in RA.

Questions and comments:

1. In the table of demographical and clinical parameters of early and advanced RA patients' all data should be separately shown for the two groups. 

2. Age would be an important factor when patient groups are compared since aging is associated with increased levels of circulating cytokines and proinflammatory markers. Were the eRA and aRA groups age-matched?

3. S100 proteins are known as biomarkers of inflammation in many diseases, and it is not specific to RA which has to be considered in the discussion.

4. there are several typing errors in the last two paragraphs of the Discussion:

"very strong network connections between the S1000A8,..."Profiler confirmed the involvement of the S100A8, S100A9, S100A11 and S1000A12 proteins", "In conclusion, our study shows that proteins S100A8, S1000A11 and S1000A12 were increased"

Author Response

Questions and comments:

  1. In the table of demographical and clinical parameters of early and advanced RA patients' all data should be separately shown for the two groups.

Table 1 has been amended accordingly.

  1. Age would be an important factor when patient groups are compared since aging is associated with increased levels of circulating cytokines and proinflammatory markers. Were the eRA and aRA groups age-matched?

In our study, the median age of patients with early RA was 45 years, which is lower than the median age of patients with advanced RA (62 years). Despite this, we obtained higher concentrations of S100 proteins in eRA patients than in aRA patients (Figure 3). This is probably due to higher disease activity in eRA patients (measured by DAS28-ESR and DAS28-CRP) and the fact that aRA patients were treated more intensively. Indeed, all patients with aRA were treated with DMARDs, and some of them additional corticosteroids, while with eRA 40% of patients were without RA treatment, therefore the level of S100 proteins may be reduced in aRA group.

Additional sentences have been included in the discussion section (lines 284-287).

  1. S100 proteins are known as biomarkers of inflammation in many diseases, and it is not specific to RA which has to be considered in the discussion.

Thank you for this comment. The sentence has been added (lines 277-279).

  1. there are several typing errors in the last two paragraphs of the Discussion:

"very strong network connections between the S1000A8,..."Profiler confirmed the involvement of the S100A8, S100A9, S100A11 and S1000A12 proteins", "In conclusion, our study shows that proteins S100A8, S1000A11 and S1000A12 were increased"

Thank you for this comment. The sentences have been rephrased (lines 317-329).

Reviewer 2 Report

This manuscript adds interesting informations to the field of monocytes and S100 proteins in RA, although the numbers of patients in the different groups are low (which could be improved for figure 5). Especially it proposes the concept of early versus advanced disease in RA.

Major comments:

Table 1: Clinical and laboratory data of early and advanced RA as well as from healthy controls should be compared in this table (in separate columns).

Did the authors perform subanalyses of S100 data from patients, as 50% of patients are treated with MTX and 30.2% with methylprednisolone (which is a really high percentage). Are there different results?

Page 3, "2.2. Viability of the monocytic THP-1 cell line treated with budesonide and RG108": This result should be summarised in the text, not only  be shown as a supplement, before the conclusion why certain concentrations were selected.

Abstract and conclusion/end of discussion: I propose "Our data suggest that the proteins S100A8 and S100A12 are strongly increased during ongoing inflammation, especially in eRA which was more potent than in aRA, so they can be used as a biomarker of disease activity." The aspect "especially in eRA which was more potent than in aRA" should be discussed independently, as this aspect may  not always depend on the disease activity alone. The sentence "In conclusion, ..." has  to be revised for better understanding.

The discussion does not include limitations of the study, although several limitations could be discussed. This should be done.

Minor comments:

Page 2, 3rd paragraph: "N-Phthalyl-L-tryptofan" should be "N-Phthalyl-L-tryptophan"

Page 8, 3rd paragraph: "because is more selective" should be "as it is ...", "RG108 has not been studied in relation to rheumatic diseases" should be "... studied  in rheumatic diseases".

Author Response

Major comments:

Table 1: Clinical and laboratory data of early and advanced RA as well as from healthy controls should be compared in this table (in separate columns).

Table 1 has been amended accordingly.

Did the authors perform subanalyses of S100 data from patients, as 50% of patients are treated with MTX and 30.2% with methylprednisolone (which is a really high percentage). Are there different results?

Sub-analyses of protein concentrations from the S100 family were also performed for groups of patients: treated with methotrexate alone, methotrexate with methylprednisolone, treated with other DMARDs and without treatment. Using ANOVA (non-parametric test) in particular the Kruskal-Wallis test, we did not observe statistical differences between these subgroups (for S100A8 p=0.29; for S100A11 p=0.55, for S100A12 p=0.079). Data not shown.

Page 3, "2.2. Viability of the monocytic THP-1 cell line treated with budesonide and RG108": This result should be summarised in the text, not only shown as a supplement, before the conclusion why certain concentrations were selected.

Thank you for this comment. The sentences have been rephrased (lines 123-126).

Abstract and conclusion/end of the discussion: I propose "Our data suggest that the proteins S100A8 and S100A12 are strongly increased during ongoing inflammation, especially in eRA which was more potent than in aRA, so they can be used as a biomarker of disease activity." The aspect "especially in eRA which was more potent than in aRA" should be discussed independently, as this aspect may  not always depend on the disease activity alone. The sentence "In conclusion, ..." has  to be revised for better understanding.

The sentences have been rephrased (lines 333-335 and lines 343-344). and

The discussion does not include limitations of the study, although several limitations could be discussed. This should be done.

The limitations of our study have been included in the discussion section (lines 333-341).

Minor comments:\]preq             `

Page 2, 3rd paragraph: "N-Phthalyl-L-tryptofan" should be "N-Phthalyl-L-tryptophan"

This has been corrected (line 65).

Page 8, 3rd paragraph: "because is more selective" should be "as it is ...", "RG108 has not been studied in relation to rheumatic diseases" should be "... studied  in rheumatic diseases".

This has been corrected (line 259).

Round 2

Reviewer 2 Report

-